# COVID-19 vaccine uptake, confidence and hesitancy in rural KwaZulu-Natal, South Africa between April 2021 and April 2022: A continuous cross-sectional surveillance study

Rachael Piltch-Loeb[1,2], Lusanda Mazibuko[3], Eva Stanton[1], Thobeka Mngomezulu[3], Dickman Gareta[3], Siyabonga Nxumalo[3], John D. Kraemer[4], Kobus Herbst[3,5], Mark J. Siedner[3,6,7,8], Guy Harling[3,9,10,11] *

1 Emergency Preparedness Research Evaluation and Practice (EPREP) Program, Division of Policy Translation and Leadership Development, Harvard T.H. Chan School of Public Health, Boston, Massachusetts, United States of America, 2 Department of Biostatistics, Harvard TH Chan School of Public Health, Boston, Massachusetts, United States of America, 3 Africa Health Research Institute, KwaZulu-Natal, South Africa, 4 Department of Health Management and Policy, Georgetown University School of Health, Washington, DC, United States of America, 5 DSI-MRC South African Population Research Infrastructure Network (SAPRIN), Durban, South Africa, 6 School of Clinical Medicine, University of KwaZulu-Natal, Durban, South Africa, 7 Division of Infectious Diseases, Massachusetts General Hospital, Boston, Massachusetts, United States of America, 8 Harvard Medical School, Boston, Massachusetts, United States of America, 9 Institute for Global Health, University College London, London, United Kingdom, 10 MRC/Wits Rural Public Health & Health Transitions Research Unit (Agincourt), University of the Witwatersrand, Johannesburg, South Africa, 11 School of Nursing & Public Health, College of Health Sciences, University of KwaZulu-Natal, KwaZulu-Natal, South Africa

* g.harling@ucl.ac.uk

**Data Availability Statement:** Data collected in the AHRI surveillance system are routinely available in

## Abstract

High COVID-19 vaccine hesitancy in South Africa limits protection against future epidemic waves. We evaluated how vaccine hesitancy and its correlates evolved April 2021-April 2022 in a well-characterized rural KwaZulu-Natal setting. All residents aged >15 in the Africa Health Research Institute's surveillance area were invited to complete a home-based, in-person interview. We described vaccine uptake and hesitancy trends, then evaluated associations with pre-existing personal factors, dynamic environmental context, and cues to action using ordinal logistic regression. Among 10,011 respondents, vaccine uptake rose as age-cohorts became vaccine-eligible before levelling off three months post-eligibility; younger age-groups had slower uptake and plateaued faster. Lifetime receipt of any COVID-19 vaccine rose from 3.0% in April-July 2021 to 32.9% in January-April 2022. Among 7,445 unvaccinated respondents, 47.7% said they would definitely take a free vaccine today in the first quarter of the study time period, falling to 32.0% in the last. By March/April 2022 only 48.0% of respondents were vaccinated or said they would definitely would take a vaccine. Predictors of lower vaccine hesitancy included being male (adjusted odds ratio [aOR]: 0.70, 95% confidence interval [CI]: 0.65–0.76), living with vaccinated household members (aOR:0.65, 95%CI: 0.59–0.71) and knowing someone who had had COVID-19 (aOR: 0.69, 95%CI: 0.59–0.80). Mistrust in government predicted greater hesitancy (aOR: 1.47, 95%CI: 1.42–1.53). Despite several COVID-19 waves, vaccine hesitancy was common in rural

pseudonymized form through https://data.ahri.org/. Data reported in this article are available from the AHRI Data Repository at https://doi.org/10.23664/AHRI.SAPRIN.COVID-19.Vaccine.Hesitancy.Dataset.

**Funding:** The African Health Research Institute's health and demographic surveillance system, which collected the data analysed, is supported by the Wellcome Trust (grant number: 201433/Z/16/Z) and the South African Department of Science and Innovation through the South African Population Research Infrastructure Network, which is hosted by the South African Medical Research Council (no grant number). GH is supported by a fellowship from the Royal Society and the Wellcome Trust (grant number: 210479/Z/18/Z). MJS is supported by the National Institutes of Health (K24 HL166024). The funders had no role in study design, data collection and analysis, decision to publish, or preparation of the manuscript.

**Competing interests:** The authors have declared that no competing interests exist.

South Africa, rising over time and closely tied to mistrust in government. However, interpersonal experiences countered hesitancy and may be entry-points for interventions.

## Introduction

Over 4 million cases and over 100,000 deaths from COVID-19 have been reported in South Africa [1]. Vaccination is likely the most effective mechanism to prevent morbidity and mortality from COVID-19, particularly in lower- and middle-income countries [2]. Although South Africa began its national COVID-19 vaccine program in May 2021, the country did not meet its goal of vaccinating two-thirds of the population by the end of 2021 [3]. Indeed, by 11 November 2022, while nearly 38 million doses of COVID-19 vaccination had been administered nationally, only 40.2% residents had received at least one dose, 35.3% a complete primary series of one or two doses, and just 6.4% a booster dose [1]. Vaccine hesitancy remains a barrier to uptake in South Africa [3–6]. A systematic review of surveys conducted up to March 2021 found COVID-19 vaccine acceptance levels of between 52% and 82%, with hesitancy growing over time [7]. Vaccine hesitancy has been associated with changes in the COVID-19 response and vaccine rollout as well as the virus itself. For example, vaccine hesitancy was particularly high in December 2020, when the AstraZeneca vaccine was reported to be ineffective against the beta variant [7].

Vaccine hesitancy is defined as "delay in acceptance or refusal of vaccines despite availability of vaccination services" [8]. To reduce the impact of vaccine hesitancy on uptake, a comprehensive understanding of its determinants is required. Vaccine hesitancy was reported as a top-ten threat to global health in 2019 [9]; its importance has been both illustrated and exacerbated by COVID-19. Vaccine hesitancy can be understood through several domains including: (i) contextual influences like communication and political environment, culture, and socioeconomic status; (ii) individual and group influences such as personal and familial experiences with vaccination, perceptions of risks and benefits, and personal experience with the medical system; and (iii) vaccine-specific influences including perceptions of vaccines' efficacy or the vaccine rollout [10]. In addition, individuals' direct experiences with the virus, and the experiences of close others, can be influential cues to action that shift perceptions and increase intervention uptake [11]. Importantly, vaccine hesitancy is a spectrum rather than a binary state, and an individual's position on this spectrum is dynamic, affected by the person's changing personal and social context. Characterizing trends in vaccine hesitancy, or conversely confidence, can help identify factors associated with increasing (and decreasing) vaccine acceptance and assess the impact of external factors (e.g., epidemic waves) and interventions (e.g., educational programs).

Within South Africa, national telephonic surveys showed that by April/May 2021 over three-quarters of respondents were willing to be vaccinated; older adults and those with pre-existing health conditions were more vaccine acceptant [12]. COVID-19 vaccine hesitancy in South Africa has been associated with several demographic factors. More vaccine hesitancy was reported among younger, less educated, and White people [13–15], but unlike in high-income settings, females report vaccine hesitancy higher than males [13], with females reporting more stigmatizing beliefs about people who have had COVID-19 [15]. Perceived risk of COVID-19 was associated with lower rates of vaccine hesitancy [13, 14], and was greater for those at greater risk of adverse outcomes from COVID-19, i.e. older people and those with chronic conditions [7]. Risk perception was also higher among those who knew someone who had had COVID-19. Nevertheless, COVID-19 risk perception may be lower in Africa than

elsewhere, based on a perception that the initial wave had limited impact on the continent [16].

These South African findings are important, however they leave critical questions unanswered. Most previous work used online opt-in methods, or telephonic interviews based on pre-consented cohorts. While such surveys offer insight into vaccine acceptance nationally, they likely underrepresent rural and lowest-income citizens, who are likely to have different hesitancy rates and predictors [13, 14]. Furthermore, past work has not systematically examined factors that operate at each level of WHO's framework on determinants of vaccine hesitancy [10]. Additionally, most past studies were conducted prior to widespread COVID-19 vaccine eligibility in mid-2021, so does not capture subsequent changes in vaccination hesitancy. We therefore focus on addressing these three critical gaps.

We analyzed data from a continuous population-wide surveillance in rural KwaZulu-Natal using potential predictors based on WHO's vaccine determinants framework. We analyzed how COVID-19 vaccine hesitancy evolved between April 2021 and April 2022 as the South African national vaccine rollout progressed during the delta and omicron waves (Fig 1A and 1B). We hypothesized that COVID-19 vaccine acceptability would: (1) increase over time, as vaccine availability and eligibility increased; (2) increase most among older adults who were both eligible soonest and most at risk from the virus; and (iii) be correlated with both environment and "cues to action".

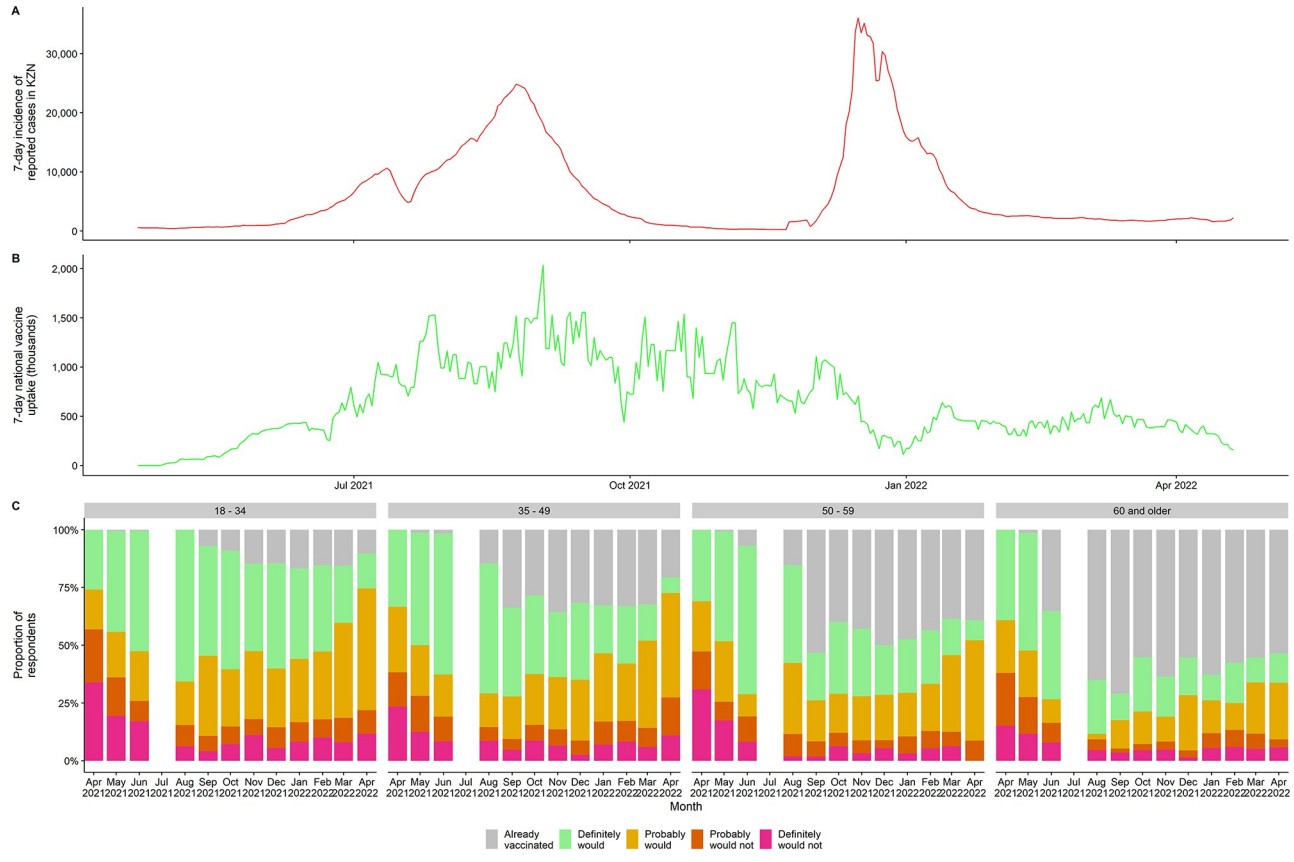

**Fig 1. Timeline of A) daily reported COVID-19 cases in KwaZulu-Natal, B) daily Vaccine uptake in South Africa, c) vaccine uptake and willingness in Hlabisa subdistrict.** Note: No in-person interviews were conducted in July in response to concern regarding rising case numbers locally.

## Methods

### Setting and survey design

The study was conducted in the Africa Health Research Institute (AHRI) Population Intervention Platform surveillance area (PIPSA) in the uMkhanyakude district of KwaZulu-Natal province. Since 2000 AHRI has conducted health and demographic surveillance of the approximately 20,000 households (totaling over 100,000 resident individuals) in the PIPSA contacting all households three times per year to interview one household senior representative, acting as proxy for all other members [17]. Since 2017, one of these visits has been in-person while the other two used a telephonic interview platform. The in-person visit additionally includes an individual health interview with any resident member aged 15 or above. All data is collected by interviewers on tablet computers using Survey Solutions and REDCap software.

Beginning in April 2020, AHRI implemented a COVID-19 surveillance protocol, shared with colleagues at other South African health surveillance sites [18]. This was initially entirely telephonic and answered by the household proxy but switched to a hybrid telephonic and in-person visits from April 2021. Within this hybrid surveillance system, we focused on the annual in-person interviews, since these included the vaccine willingness question. We linked individual responses to relevant household-level in-person proxy responses conducted in close temporal proximity (typically within a few days). Our analysis period covers first round of in-person interviews post-COVID (20 April 2021 to 19 April 2022).

### Outcomes

Willingness to be vaccinated was characterized based on responses to two questions: first, "Has this person ever been vaccinated against COVID-19?", and if the answer to this was negative and the relevant person was present, they were asked "If a COVID-19 vaccine was available to you right now at no cost, would you agree to be vaccinated?" Respondents could reply on a 4-point Likert scale (options "definitely would", "probably would", "probably would not", "definitely would not". We examined three outcomes based on responses to these questions.

Our primary outcome was a five-level variable, starting from "already vaccinated" and ending with "definitely would not get vaccinated". We also considered dichotomized outcomes: (i) "already vaccinated", "definitely would" and "probably would" versus any negative response; (ii) "definitely would not" vs. all other response categories.

### Independent variables

Based on our conceptual model we anticipated several factors would predict vaccine acceptance (S1 Table). In addition to sex and age (four categories based on vaccine eligibility date: 18–34, 35–49, 50–59, 60+) we also considered contextual factors, group/community factors, and cues to action. Personal contextual factors included sources of COVID-19 information, level of mistrust in government, educational attainment and household urbanicity. Group/community influences included perceived changes in household and community economic wellbeing during the pandemic, household composition (any member aged >60) and COVID-related stereotype and anticipated stigma (composite scores from six- and five-item scales respectively, adapted from [19]). Finally, proximal cues to action included respondents' level of concern if they acquired COVID, knowledge of others with COVID, vaccination behavior of other household members, self-reported mental health (using the PHQ4 measure [20]) and current provincial epidemic situation (reported cases in past seven days in KwaZulu-Natal).

**Statistical analyses.** From the initial sample of all in-person interviews (n = 15,755), we dropped observations missing an answer for the vaccine uptake or willingness questions (e.g.,

refused to respond or said, "don't know", n = 46, 0.3%). We further dropped all individuals without linkable household interviews (n = 4634, 29.5%) due to non-contact or non-consent from the household proxy respondent) or missing information on educational attainment (n = 929, 8.4% of remaining interviews). Given low levels of item non-response (n = 135, 1.3% of remaining interviews incomplete) we conducted a complete case analysis by dropping observations with missing item values. After describing our sample using appropriate measures of centrality and dispersion, we evaluated how dropped and retained individuals differed on key variables and ran bivariate regressions for vaccine hesitancy and all independent variables.

We then described vaccine acceptance trends over time, plotting five-level vaccine acceptability across the study period and the proportion of all unvaccinated individuals in each age/sex group who were uncertain (probably would or probably would not vaccinate). We next ran regression models first adding age and sex, and then variables for context, environment, and cues to action. We visualized these results by plotting predicted probabilities of being in each category given respondent values for key independent variables. Finally, we conducted sensitivity analyses by rerunning our regressions using binary logistic models for our alternative binary outcome measures (any negative response and definite negative response). All regression models included community-level random effects (n = 24) to account for clustering of outcomes. All statistical analyses were conducted in Stata 15.1. All data underlying our analyses are publicly available on request [21].

**Ethical considerations.**   All households had previously given consent to be contacted by phone and each respondent gave written consent (or if aged under 18, written assent following written consent from a parent or guardian). Study procedures were approved by AHRI's community advisory board, KwaZulu-Natal's provincial Department of Health Research Ethics Committees (REC) and the University of KwaZulu-Natal's Biomedical REC.

**Inclusivity in global research.**   Additional information regarding the ethical, cultural, and scientific considerations specific to inclusivity in global research is included as S1 Text.

## Results

After all exclusions, our analytical sample was 10,011 (Table 1). The sample was majority female (67.1%) and skewed towards younger adults (61.9% under age 50), in line with local demographics. A minority lived with someone else who has been vaccinated (27.6%), even though 55.4% lived with an adult over age 60. Almost everyone relied on traditional information sources for COVID-19 content (98.3%, of which the great majority used radio or television, with only 7.3% using newspapers and 1.1% government websites), with a minority (15.7%) also using healthcare sources. Very few respondents knew someone who had had COVID-19 (6.3%), and a majority were not at all (21.8%) or slightly (38.8%) concerned about future COVID-19 infection.

Vaccine hesitancy status varied widely; 25.6% (95% confidence interval [CI]: 24.7–26.5%) were already vaccinated, 32.6% (95%CI: 31.7–33.5%) said they would definitely take a vaccine, 24.4% (95%CI: 23.5–25.2%) that they probably would, 8.7% (95%CI: 8.2–9.3%) that they probably would not and 8.7% (95%CI: 8.1–9.3%) that they definitely would not. Our retained sample was broadly similar to all respondents but was slightly younger, more educated, less likely to know someone who had COVID-19, more concerned about the virus, had higher PHQ-4 scores and responded when COVID-19 case counts were lower (S3 Table).

Vaccine uptake timing largely reflected when different ages became eligible (60+: 17 May; 50–59: 1 July; 35–49: 1 August; 18–34: 1 September), although healthcare workers and other essential groups had earlier access. At each age, eligibility led to rapid initial uptake in

**Table 1. Descriptive statistics by vaccine uptake/hesitancy.**

| | Total | Already vaccinated | Definitely would | Probably would | Probably would not | Definitely would not | p-value |
|---|---|---|---|---|---|---|---|
| N | 10,011 | 2,562 | 3,267 | 2,442 | 872 | 868 | |
| Female | 67.1% | 77.2% | 65.5% | 62.5% | 62.7% | 61.4% | <0.001 |
| Age group | | | | | | | <0.001 |
| 18–34 | 37.4% | 14.3% | 45.8% | 45.0% | 42.4% | 47.0% | |
| 35–49 | 24.5% | 24.0% | 24.1% | 25.8% | 25.2% | 23.2% | |
| 50–59 | 14.7% | 19.4% | 13.1% | 13.6% | 12.2% | 12.1% | |
| 60+ | 23.4% | 42.3% | 17.0% | 15.5% | 20.2% | 17.7% | |
| COVID-19 information sources | | | | | | | |
| Traditional | 98.3% | 98.1% | 98.8% | 97.7% | 98.5% | 98.0% | 0.020 |
| Personal network | 4.7% | 4.4% | 4.3% | 5.2% | 4.9% | 5.8% | 0.22 |
| Healthcare | 15.7% | 17.6% | 12.4% | 18.1% | 15.4% | 16.5% | <0.001 |
| Community | 3.6% | 2.6% | 3.9% | 3.7% | 4.4% | 5.0% | 0.006 |
| Mistrust in government (raw score) | 6 (3–7) | 6 (3–6) | 6 (3–7) | 6 (5–8) | 6 (6–9) | 7 (6–9) | <0.001 |
| Highest educational attainment | | | | | | | <0.001 |
| None | 11.2% | 16.7% | 9.2% | 9.0% | 10.8% | 9.8% | |
| Primary | 16.2% | 21.2% | 14.7% | 14.0% | 14.3% | 15.3% | |
| Some secondary | 42.5% | 32.7% | 46.0% | 45.5% | 47.4% | 44.9% | |
| Completed secondary | 23.3% | 20.5% | 24.2% | 24.9% | 22.2% | 24.4% | |
| Any tertiary | 6.8% | 8.9% | 5.8% | 6.7% | 5.3% | 5.5% | |
| Urbanicity of household | | | | | | | <0.001 |
| Peri-Urban | 24.6% | 26.0% | 25.9% | 23.1% | 21.3% | 22.8% | |
| Rural | 67.4% | 64.0% | 65.6% | 69.5% | 72.5% | 72.9% | |
| Urban | 8.0% | 10.0% | 8.5% | 7.4% | 6.2% | 4.3% | |
| Change in economic stability | | | | | | | <0.001 |
| Much better off | 0.6% | 0.5% | 0.4% | 0.5% | 1.5% | 0.5% | |
| A little better off | 3.6% | 2.8% | 3.4% | 5.0% | 2.3% | 4.1% | |
| About the same | 79.6% | 81.7% | 78.1% | 77.4% | 82.9% | 81.6% | |
| A little worse off | 7.9% | 6.0% | 8.8% | 8.1% | 7.8% | 9.0% | |
| Much worse off | 8.4% | 9.0% | 9.2% | 9.0% | 5.5% | 4.8% | |
| Change in community wellbeing | | | | | | | 0.006 |
| Got better | 1.7% | 2.0% | 1.5% | 1.9% | 1.1% | 2.0% | |
| Stayed the same | 80.0% | 81.6% | 78.7% | 79.0% | 80.3% | 83.1% | |
| Got worse | 18.2% | 16.4% | 19.7% | 19.2% | 18.6% | 15.0% | |
| Has household member aged 60+ | 55.4% | 62.9% | 52.9% | 51.4% | 56.8% | 52.4% | <0.001 |
| COVID stereotype stigma score | 6 (6–6) | 6 (6–6) | 6 (6–6) | 6 (6–6) | 6 (6–6) | 6 (6–6) | <0.001 |
| COVID anticipated stigma score | 5 (3–5) | 5 (3–5) | 5 (3–5) | 5 (2–5) | 5 (3–5) | 5 (3–5) | <0.001 |
| Future COVID infection concern level | | | | | | | <0.001 |
| Not at all | 21.8% | 21.2% | 26.2% | 16.2% | 17.1% | 27.5% | |
| Slightly concerned | 38.8% | 41.0% | 27.1% | 55.3% | 32.6% | 35.7% | |
| Moderately concerned | 16.8% | 15.7% | 13.7% | 19.2% | 22.9% | 18.3% | |
| Very concerned | 22.7% | 22.2% | 32.9% | 9.3% | 27.4% | 18.4% | |
| Knows someone who has had COVID | 6.3% | 7.6% | 6.2% | 5.6% | 5.6% | 5.6% | 0.030 |
| Any other household members vaccinated | 27.6% | 39.5% | 23.0% | 27.1% | 21.2% | 17.3% | <0.001 |
| PHQ-4 categories | | | | | | | <0.001 |
| Normal | 66.2% | 65.5% | 70.0% | 61.1% | 69.4% | 65.4% | |
| Mild | 19.7% | 16.5% | 18.1% | 25.6% | 16.6% | 21.2% | |

(*Continued*)

**Table 1.** (Continued)

| | Total | Already vaccinated | Definitely would | Probably would | Probably would not | Definitely would not | p-value |
|---|---|---|---|---|---|---|---|
| Moderate | 11.4% | 15.5% | 8.9% | 11.7% | 11.8% | 7.8% | |
| Severe | 2.7% | 2.4% | 3.0% | 1.6% | 2.2% | 5.5% | |
| Past 7-day KZN cases per1000 population | 0.15 (0.06–0.22) | 0.17 (0.08–0.23) | 0.14 (0.06–0.24) | 0.16 (0.07–0.21) | 0.14 (0.06–0.20) | 0.09 (0.05–0.19) | <0.001 |

Note: Figures for categorical variables are row percentages with $\chi_4^2$ tests of difference; figures for continuous variables are median and (interquartile range) with Kruskall-Wallis tests of difference.

the first 2–3 months, followed by a leveling-off; younger age groups saw less substantial initial uptake and more rapid plateauing (Fig 1C). Among those not yet vaccinated, the proportion saying they definitely or probably would not take a vaccine dropped from April to June 2021 and remained below 25% for almost all age-group months thereafter. The proportion saying they definitely would get vaccinated fell over time, reflecting both increased vaccination, and a shift towards increasingly saying they probably would get vaccinated. Initial vaccination increases occurred in the context of South Africa's Delta wave which peaked in KZN in September 2021, however the later Omicron wave did not lead to a parallel increase in vaccine uptake, despite greater case numbers (Fig 1A and 1B). Vaccine uptake rates remained below 2021 rates both nationally and locally throughout the four months of 2022 covered by this study.

Vaccine uncertainty, defined as respondents who stated they probably would or probably would not accept a vaccine, was relatively stable over time as a proportion of the whole population, although represented an increasing proportion of those not yet vaccinated (Fig 2). Similarly, uncertainty was greatest in younger age-groups where more people are not yet vaccinated and among males; these differences were not present when looking only at those not yet vaccinated.

Detailed monthly values for variables other than uptake and willingness are provided in S2 Table. Most characteristics were stable across months, with a few exceptions (e.g., urbanicity varied as interviewers moved sequentially through areas over time). In later interview months, respondents were more likely to report knowing someone who had been vaccinated, consistent with the South African vaccination campaign, and less likely to report concern about contracting COVID-19.

In terms of pre-existing factors, vaccine hesitant individuals were more likely to: be male; be younger; use community/personal network, but not healthcare, information sources; mistrust government; have a secondary education (vs none, primary or tertiary); and live rurally (Table 1). These patterns persisted in multivariable analysis of socio-demographics, aside from urbanicity becoming null and tertiary education becoming less protective (Table 2). Contextually, hesitant individuals were more likely to be economically worse-off since COVID-19, not to cohabit with older persons; and to report more COVID-19 related stigma. All these patterns remained consistent in a multivariable model of contextual factors. Among cues to action, more hesitant individuals were less concerned about COVID-19 exposure, more likely to report not knowing someone who had had COVID-19, and less likely to live with anyone who had been vaccinated. In multivariable analysis of cues, knowing COVID-19 cases and having other household members vaccinated remained protective, but concern with catching COVID-19 and current epidemic severity were inconsistent. Vaccine hesitancy declined over time in all regression models. In a fully adjusted model, most factors remained unchanged,

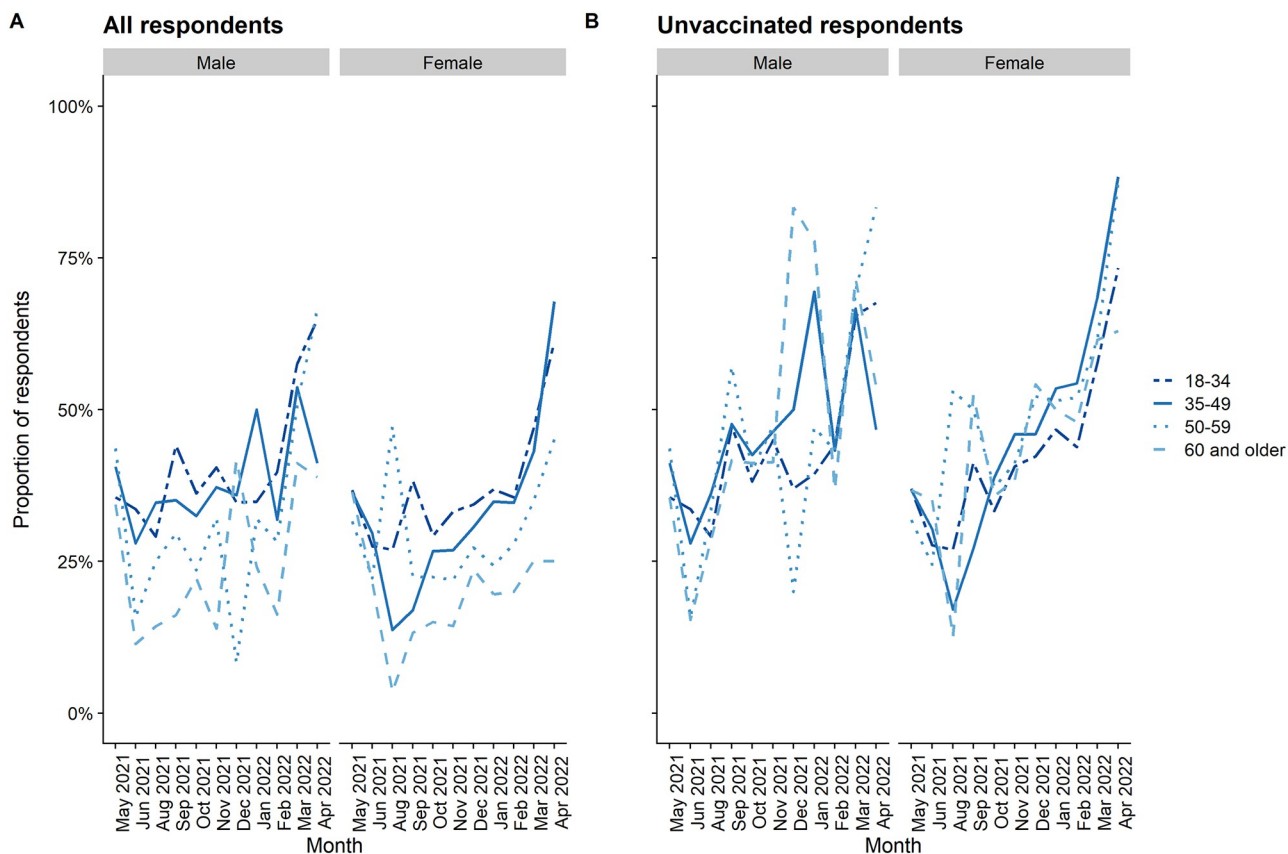

**Fig 2. Proportion of respondents uncertain regarding future vaccine willingness: A) among all respondents, B) among unvaccinated respondents.**

although more education became consistently protective and having older household members became non-significant.

Vaccine hesitancy was more patterned by age than sex, with a relatively consistent proportion of all ages saying they definitely would get vaccinated but diverging trends among those already vaccinated and those displaying any hesitancy (Fig 3A). A similar pattern was seen for government mistrust (Fig 3C). Having other vaccinated household members and knowing COVID-19 cases affected vaccine uptake more than strength of hesitancy (Fig 3D and 3E). The vaccinated were most likely to report very high and low levels of concern about COVID-19 exposure (Fig 3F), suggesting that some who are motivated to vaccinate remained worried about infection while others were reassured. Our sensitivity analyses using any negative response or definite negative response as the outcome did not show substantively differing results (S4 Table).

## Discussion

Using over 10,000 responses from population-wide surveillance in rural KwaZulu-Natal, South Africa we showed how COVID-19 vaccine hesitancy evolved in the midst of the pandemic from April 2021 to April 2022. By mapping survey questions onto the WHO's vaccine determinants framework, we evaluated which factors predicted these changes. We found slowing vaccine uptake and rising hesitancy among the unvaccinated, with both close personal experiences, such as knowing someone who had COVID-19, and broader societal perceptions,

**Table 2. Multivariable regression models of vaccine hesitancy (five levels).**

| | Demographics | | Existing | | Contextual | | Cues to action | | All variables | |
|---|---|---|---|---|---|---|---|---|---|---|
| Female vs Male | 0.70 | [0.65, 0.76] | | | | | | | 0.70 | [0.65, 0.76] |
| Age (vs 18–34 year olds) | 1.00 | | | | | | | | 1.00 | |
| 35–49 years | 0.65 | [0.60, 0.72] | | | | | | | 0.65 | [0.59, 0.72] |
| 50–59 years | 0.47 | [0.42, 0.52] | | | | | | | 0.42 | [0.37, 0.47] |
| 60 years and older | 0.27 | [0.24, 0.30] | | | | | | | 0.21 | [0.19, 0.25] |
| COVID information sources (yes vs no for each) | | | | | | | | | | |
| Traditional | | | 0.95 | [0.71, 1.26] | | | | | 0.88 | [0.66, 1.18] |
| Personal network | | | 1.17 | [0.99, 1.39] | | | | | 1.06 | [0.89, 1.26] |
| Healthcare | | | 1.05 | [0.95, 1.16] | | | | | 1.09 | [0.98, 1.20] |
| Community | | | 1.18 | [0.98, 1.43] | | | | | 1.15 | [0.95, 1.40] |
| Mistrust in government, z-score | | | 1.50 | [1.44, 1.55] | | | | | 1.47 | [1.42, 1.53] |
| Highest educational attainment (vs None) | | | 1.00 | | | | | | 1.00 | |
| Primary | | | 1.19 | [1.03, 1.38] | | | | | 0.91 | [0.78, 1.05] |
| Some secondary | | | 2.05 | [1.81, 2.33] | | | | | 0.78 | [0.67, 0.91] |
| Completed secondary | | | 1.93 | [1.68, 2.21] | | | | | 0.69 | [0.59, 0.82] |
| Any tertiary | | | 1.44 | [1.20, 1.73] | | | | | 0.63 | [0.51, 0.77] |
| Household location (vs rural) | | | 1.00 | | | | | | 1.00 | |
| Peri-Urban | | | 1.13 | [0.99, 1.29] | | | | | 1.12 | [0.98, 1.27] |
| Urban | | | 0.87 | [0.64, 1.19] | | | | | 0.96 | [0.74, 1.25] |
| Household economic change (vs None) | | | | | 1.00 | | | | 1.00 | |
| Much better off | | | | | 1.33 | [0.82, 2.15] | | | 1.13 | [0.69, 1.83] |
| A little better off | | | | | 1.17 | [0.97, 1.42] | | | 1.08 | [0.89, 1.31] |
| A little worse off | | | | | 0.99 | [0.86, 1.13] | | | 1.00 | [0.87, 1.15] |
| Much worse off | | | | | 0.89 | [0.76, 1.03] | | | 0.90 | [0.77, 1.05] |
| Community wellbeing change (vs None) | | | | | 1.00 | | | | 1.00 | |
| Got better | | | | | 0.86 | [0.65, 1.14] | | | 0.99 | [0.75, 1.31] |
| Got worse | | | | | 1.06 | [0.95, 1.18] | | | 1.06 | [0.95, 1.19] |
| Household member aged 60+ | | | | | 0.69 | [0.64, 0.74] | | | 1.04 | [0.95, 1.13] |
| COVID stereotype stigma, z-score | | | | | 1.04 | [0.99, 1.09] | | | 1.01 | [0.96, 1.06] |
| COVID anticipated stigma, z-score | | | | | 1.05 | [1.01, 1.09] | | | 1.02 | [0.97, 1.06] |
| KZN cases per 1000 pop in past 7 days | | | | | | | 0.86 | [0.57, 1.28] | 0.80 | [0.53, 1.21] |
| Concern if got COVID (vs Not at all) | | | | | | | 1.00 | | 1.00 | |
| Slightly concerned | | | | | | | 1.31 | [1.18, 1.45] | 1.35 | [1.22, 1.50] |
| Moderately concerned | | | | | | | 1.49 | [1.32, 1.69] | 1.38 | [1.21, 1.56] |
| Very concerned | | | | | | | 0.79 | [0.70, 0.88] | 0.84 | [0.75, 0.94] |
| Knows someone who has had COVID | | | | | | | 0.74 | [0.64, 0.86] | 0.69 | [0.59, 0.80] |
| Any other household members vaccinated | | | | | | | 0.72 | [0.66, 0.79] | 0.65 | [0.59, 0.71] |
| PHQ-4 (vs normal) | | | | | | | 1.00 | | 1.00 | |
| Mild | | | | | | | 0.92 | [0.84, 1.02] | 0.96 | [0.87, 1.06] |
| Moderate | | | | | | | 0.71 | [0.63, 0.80] | 0.91 | [0.80, 1.03] |
| Severe | | | | | | | 1.05 | [0.84, 1.33] | 1.16 | [0.91, 1.47] |
| Interview Month (vs April) | 1.00 | | 1.00 | | 1.00 | | 1.00 | | 1.00 | |
| May 2021 | 0.53 | [0.42, 0.68] | 0.60 | [0.46, 0.77] | 0.55 | [0.43, 0.71] | 0.52 | [0.41, 0.67] | 0.59 | [0.45, 0.76] |
| June 2021 | 0.26 | [0.20, 0.34] | 0.39 | [0.30, 0.51] | 0.33 | [0.25, 0.43] | 0.34 | [0.26, 0.45] | 0.35 | [0.26, 0.46] |
| August 2021 | 0.16 | [0.12, 0.22] | 0.26 | [0.19, 0.35] | 0.22 | [0.16, 0.30] | 0.27 | [0.15, 0.52] | 0.27 | [0.14, 0.51] |
| September 2021 | 0.13 | [0.10, 0.17] | 0.18 | [0.13, 0.25] | 0.17 | [0.13, 0.23] | 0.18 | [0.13, 0.26] | 0.17 | [0.12, 0.23] |
| October 2021 | 0.15 | [0.12, 0.20] | 0.24 | [0.18, 0.32] | 0.19 | [0.14, 0.25] | 0.19 | [0.14, 0.25] | 0.21 | [0.16, 0.28] |

(*Continued*)

**Table 2.** (Continued)

| | Demographics | | Existing | | Contextual | | Cues to action | | All variables | |
|---|---|---|---|---|---|---|---|---|---|---|
| November 2021 | 0.14 | [0.11, 0.18] | 0.25 | [0.19, 0.32] | 0.17 | [0.13, 0.23] | 0.17 | [0.13, 0.23] | 0.22 | [0.16, 0.28] |
| December 2021 | 0.13 | [0.10, 0.18] | 0.22 | [0.16, 0.30] | 0.16 | [0.11, 0.21] | 0.17 | [0.12, 0.24] | 0.21 | [0.15, 0.31] |
| January 2022 | 0.15 | [0.11, 0.20] | 0.26 | [0.19, 0.35] | 0.18 | [0.14, 0.25] | 0.19 | [0.14, 0.27] | 0.24 | [0.17, 0.34] |
| February 2022 | 0.15 | [0.12, 0.20] | 0.22 | [0.17, 0.30] | 0.17 | [0.13, 0.23] | 0.18 | [0.13, 0.23] | 0.22 | [0.17, 0.29] |
| March 2022 | 0.21 | [0.16, 0.27] | 0.33 | [0.25, 0.43] | 0.27 | [0.21, 0.35] | 0.26 | [0.20, 0.34] | 0.29 | [0.22, 0.38] |
| April 2022 | 0.32 | [0.23, 0.44] | 0.51 | [0.37, 0.71] | 0.45 | [0.33, 0.63] | 0.44 | [0.31, 0.61] | 0.42 | [0.30, 0.58] |
| Community level variance (random effect) | 0.02 | | 0.03 | | 0.03 | | 0.03 | | 0.02 | |

N = 10,011. All models are two-level (individuals nested in communities) random effects order logistic regressions. Higher values represent greater hesitancy.

such as distrust of government, as key predictors of hesitancy. This analysis is one of the first to examine of COVID-19 vaccine hesitancy in longitudinal, population-level data in sub-Saharan Africa, and particularly to evaluate determinants of behavior and attitudes [22, 23].

Vaccine uptake increased over time as vaccine eligibility expanded from only frontline healthcare workers in April to all adults from September 2021. However, despite vaccine eligibility, consistent availability and the increasing normality of COVID-19 vaccination, hesitancy also increased over time among the unvaccinated. Notably, even as vaccine uptake fell from November 2021 onwards, vaccine hesitancy continued to increase, suggesting a possible

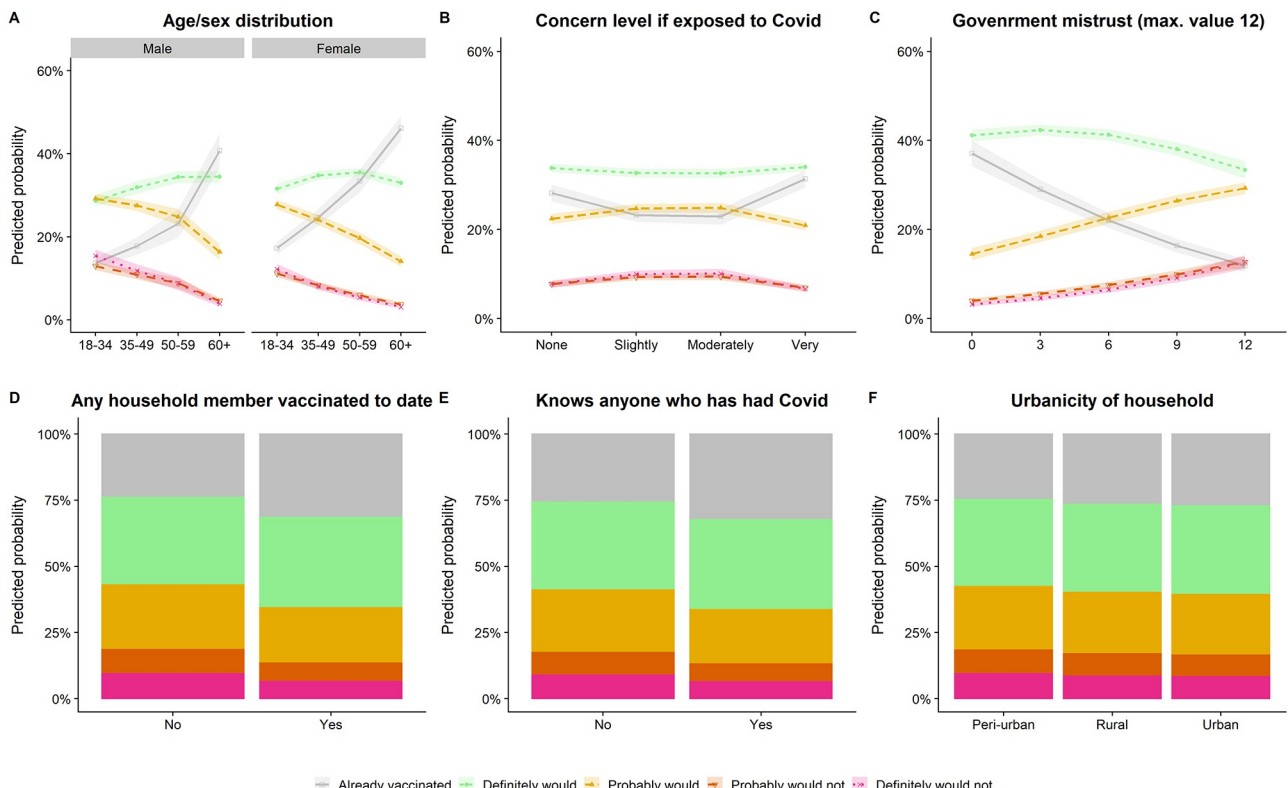

**Fig 3. Predicted probabilities from multivariable models for five levels of vaccine hesitancy for key variables, adjusting for other model covariates.**

'hardening' of hesitancy over time. By April 2022 only 48% of respondents aged 50–59, and substantially fewer younger adults, reported they had been or definitely would get vaccinated. Moreover, there was a shift over time from respondents stating that they 'definitely would' take a free COVID-19 vaccine today to stating that they 'probably would'. Low vaccine uptake is in line with falling uptake rates over time and greater vaccine hesitancy among younger adults nationally [3, 12]. In our setting low uptake in part reflects its young demographics, while increasing hesitancy may reflect reduced threat perception, increasing vaccine misinformation, or other factors.

In seeking to understand these low vaccine uptake rates, this study identified two strong positive proximal predictors of vaccine acceptance: living in households where others had already been vaccinated, and knowing someone who had had COVID-19—a relative rarity in the setting. The household clustering of vaccine acceptance aligns with evidence from a South African national telephone survey in the same period [24], while a hesitancy-reducing effect of knowing someone with COVID-19 (especially severe cases) has been seen elsewhere including in Africa [25–27]. Notably, the effect of living with vaccinated household members did not appear to be a function of co-habitation with people vulnerable to COVID-19, since this association was independent of household member age. Potential mechanisms driving these results include reduced vaccine-related fear after seeing a family member successfully vaccinated and increased infection-related fear after seeing close others infected. The latter mechanism is supported by the lower hesitancy expressed in interviews conducted when provincial case counts were high. Vaccinated household members may also act more proactively as pro-vaccine advocates or dispel misinformation shared from more distal sources—as seen for influenza vaccine uptake in South Africa in 2017 [28]. This latter mechanism also suggests that supporting already-vaccinated individuals to act as ambassadors may help their families to take advantage of vaccination opportunities.

We also identified that low trust in government relating to COVID-19—a more impersonal, distal contextual factor—strongly predicted vaccine hesitancy. This reflects government's role as a vaccine proponent and distributor and aligns with other South African data showing strong associations of government mistrust with vaccine non-uptake and hesitancy [6]. Linked to government mistrust, respondents in our survey who obtained COVID-19 information from social media, friends and family or community members were more likely to be hesitant than those who did not—an association that was much attenuated once mistrust was added to the model. This aligns with other evidence of a positive association between social media usage and vaccine hesitancy globally [29]. Perhaps most surprisingly, both stereotype and perceived COVID-19 stigma were weakly associated with more hesitancy—although stigma levels were much lower in our rural survey compared to urban KwaZulu-Natal [19].

In addition to these dynamic factors, we found pre-existing socio-demographic factors were associated with vaccine hesitancy. In contrast to higher-income settings [29, 30], men in our study were more vaccine hesitant. The wider South African picture appears mixed, with several studies finding no difference between men and women [7, 12], suggesting the need for additional work to understand how mechanisms may differ by gender. Notably, a recent US study found that such that low socio-economic status was associated with increased hesitancy for women but not for men [31]—although such a pattern would not explain our findings in a setting where poverty is pervasive. As seen worldwide, younger respondents were more hesitant than older ones, potentially reflecting lower risk perceptions; however, our associations were no weaker in fully adjusted models than in bivariate ones, suggesting that we have not yet captured the relevant mechanisms in our models. Finally, vaccine hesitancy was higher in those with more education in unadjusted models, but this appears to reflect confounding by age since this association was reversed in fully adjusted models. It is unclear whether this latter

association reflects less access to valid vaccine information, less capacity to parse true and false information or less ability to access vaccines in the local area.

### Limitations

Our study has important limitations. Our focus on a single geographic area makes generalization to the wider South African population tenuous, however it also allows us to provide a whole-population view and capture key social processes through household data linkage. Non-response at the household level was low due to telephonic and in-person follow-up on a well-characterized census, minimizing the risk of participation bias within the study area; however, we were not able to interview all resident adults in the area, potentially generating new biases if non-contacted individuals were systematically different. Social desirability biases are a risk for self-reported answers relating to vaccine willingness and uptake, although the direction of these biases will depend on perceived social norms in the area. We note that some of the survey items, e.g., those relating to mistrust in government and Covid-related stigma, were not fully validated given the speed with which they were developed in the pandemic; further work is needed to validate and replicate the associations seen for these parameters. We recognize there may be other factors that impact vaccine hesitancy which we did not include in our questionnaire. For example, our study did not include survey items related to social media usage or specific pieces of mis/disinformation the user may have encountered.

### Conclusion

In a 12-month continuous surveillance program in rural KwaZulu-Natal, South Africa, we found that COVID-19 vaccination hesitancy was positively associated with broad perceptions, such as mistrust in government and COVID-related stigma, and countered by personal experiences, notably knowing others who had had COVID-19 or already taken a vaccine and living with older and thus more vulnerable individuals. These findings suggest that efforts to increase vaccine uptake—in a setting where almost half of adults remain unprotected—may require both counteracting negative external influences and highlighting how close connections could benefit from individuals' vaccination decisions. Our findings should be relevant for vaccination efforts including future boosters, influenza, and other vaccine rollouts.

### Supporting information

**S1 Text. Inclusivity in global research.**
(DOCX)

**S1 Table. Independent variable definitions.**
(DOCX)

**S2 Table. Comparison of covariate values for retained (n = 10,011) and dropped (n = 5698) individuals.**
(DOCX)

**S3 Table. Descriptive statistics for the sample by month of interview.**
(DOCX)

**S4 Table. Sensitivity analysis multivariable regressions.**
(DOCX)

## Acknowledgments

We acknowledge the efforts of all research staff working to collect these data and participants for their ongoing willingness to answer our questions.

## Author Contributions

**Conceptualization:** Rachael Piltch-Loeb, Mark J. Siedner, Guy Harling.

**Data curation:** Lusanda Mazibuko, Thobeka Mngomezulu, Dickman Gareta, Siyabonga Nxumalo, Guy Harling.

**Formal analysis:** Rachael Piltch-Loeb, Lusanda Mazibuko, John D. Kraemer, Guy Harling.

**Funding acquisition:** Kobus Herbst, Mark J. Siedner.

**Methodology:** John D. Kraemer, Kobus Herbst.

**Project administration:** Thobeka Mngomezulu.

**Supervision:** Guy Harling.

**Writing – original draft:** Rachael Piltch-Loeb, Eva Stanton.

**Writing – review & editing:** Lusanda Mazibuko, Eva Stanton, Thobeka Mngomezulu, Dickman Gareta, Siyabonga Nxumalo, John D. Kraemer, Kobus Herbst, Mark J. Siedner, Guy Harling.

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
