## [Decision Letter · Decision Letter 0]

10 Apr 2023

PGPH-D-23-00155

COVID-19 vaccine uptake, confidence and hesitancy in rural KwaZulu-Natal, South Africa between April 2021 and April 2022: a continuous cross-sectional surveillance study

Dear Dr. Guy Harling,

Thank you for submitting your manuscript to PLOS Global Public Health. After careful consideration, we feel that it has merit but does not fully meet PLOS Global Public Health’s publication criteria as it currently stands. Therefore, we invite you to submit a revised version of the manuscript that addresses the points raised during the review process listed below.

We look forward to receiving your revised manuscript.

Kind regards,

Thu-Anh Nguyen

Academic Editor

Journal Requirements:

1.  Please include the following request in the decision letter, and ping me (hlandenmark@plos.org) with follow up. “Please include a complete copy of PLOS’ questionnaire on inclusivity in global research in your revised manuscript. Our policy for research in this area aims to improve transparency in the reporting of research performed outside of researchers’ own country or community. The policy applies to researchers who have travelled to a different country to conduct research, research with Indigenous populations or their lands, and research on cultural artefacts. The questionnaire can also be requested at the journal’s discretion for any other submissions, even if these conditions are not met.  Please find more information on the policy and a link to download a blank copy of the questionnaire here: https://journals.plos.org/plosone/s/best-practices-in-research-reporting. Please upload a completed version of your questionnaire as Supporting Information when you resubmit your manuscript.

**Reviewers' comments:**

Reviewer's Responses to Questions

**Comments to the Author**

1. Does this manuscript meet PLOS Global Public Health’s publication criteria? Is the manuscript technically sound, and do the data support the conclusions? The manuscript must describe methodologically and ethically rigorous research with conclusions that are appropriately drawn based on the data presented.

Reviewer #1: Yes

Reviewer #2: Yes

2. Has the statistical analysis been performed appropriately and rigorously?

Reviewer #1: No

Reviewer #2: Yes

3. Have the authors made all data underlying the findings in their manuscript fully available (please refer to the Data Availability Statement at the start of the manuscript PDF file)?

Reviewer #1: Yes

Reviewer #2: Yes

4. Is the manuscript presented in an intelligible fashion and written in standard English?

Reviewer #1: Yes

Reviewer #2: Yes

5. Review Comments to the Author

Reviewer #1: The authors have conducted a semi- longitudinal study to assess the COVID 19 vaccine uptake,confidence and hesitancy in rural Kwazulu-Natal, South Africa.From an initial sample of 15755 persons surveyed a sub-sample of 10 011 was retained for analyses after eliminating subjects who did not fully comply with the inclusion criteria.The WHO recommended determinants/drivers of Vaccine hesitancy were interrogated ,and these included, age gender,level of education, socioeconomic status etc. The authors stated that they had performed regression analysis on their data without specifying the software employed, which would make verification difficult. The results obtained were generally in agreement with those obtained in South Africa and other African countries, although vaccine acceptance was relatively higher in South Africa than in most of Sub-Saharan Africa.The main drivers of vaccine hesitancy were mistrust of the Government, followed by low level of education and being male. Curiously enough the authors did not explore the role of social media anti-vaccine mis/disinformation which was reported as a major vaccine hesitancy driver in many studies including some of those cited by the authors of this study.

I am also concerned that the results are presented mainly as three crowded tables full of raw data making reading rather difficult. The authors should consider using histograms and pie diagrams in addition to the tables to illustrate some of their findings. In conclusion, I am satisfied that this study that was focused on a rural sample will complement the findings of other studies in urban areas.I therefore recommend that the paper be accepted subject to satisfactorily addressing the issues that I have raised above.

Reviewer #2: This article describes the findings from a cross-sectional surveillance study evaluating COVID-19 vaccine update, confidence, and hesitancy in a defined region in KwaZulu-Natal, South Africa. The authors were able to utilize an existing survey infrastructure to gather additional data about COVID-19 vaccine hesitancy and accumulate a large sample size of over 10,000 responses. They identified key contributors to vaccine hesitancy in this region, including age of individuals and proximity to others with experience with COVID-19 vaccination and/or COVID-19 disease.

Overall the manuscript is quite strong, with sample size providing sufficient statistical power to interrogate individual parameters as well as overall trends. The utilization of the WHO framework for addressing vaccine hesitancy enables the results to be compared to other analyses. The use of a pre-existing (prior to COVID) survey cohort is especially interesting as it lends validity to the interpretation of results in terms of the independence of the participants from actual vaccination clinics or healthcare settings, as well as the (presumed) established trust between those responding to the surveys and those administering them. The inclusion of a wide range of ages and level of education also lend broader interpretability of the data, though the restriction to this one location is acknowledged by both the authors and this reviewer. The authors do a good job of placing their findings in context and relating to the current literature on vaccine hesitancy.

There are no major issues to report.

A few minor issues:

(1) A few of the survey questions were structured in a way that could be considered leading. As an examples, the three Mistrust in government questions were phrased in a decidedly anti-government way. However, given the large literature on the mistrust of government of individuals in the population that contributed responses for this analysis I do not believe this directly influenced the results in a meaningful way. I could not easily locate the source of these questions in the sources for the instruments referenced to see if this had previously been addressed.

(2) Given the finding that distrust in the government was associated with hesitancy, and that government websites were included among the Traditional Sources of information, which was the category that received an overwhelmingly high response to the question "Where do you get information about COVID-19 that you trust?", the analysis may benefit from separating out government websites to a different category, or at least providing percentages of response for the different subtypes (newspapers, radio, TV, government websites) to see the contribution the government websites made to that category.

(3) There are a few missing words in the abstract and introduction sections, though I didn't see any or as many in the remaining section.

Overall this is a very interesting study with informative results that will help inform the rollout of future vaccination campaigns, either for COVID-19 or other vaccines, and possibly other health interventions.

6. PLOS authors have the option to publish the peer review history of their article (what does this mean?). If published, this will include your full peer review and any attached files.

**Do you want your identity to be public for this peer review?** For information about this choice, including consent withdrawal, please see our Privacy Policy.

Reviewer #1: No

Reviewer #2: No

---

## [Editor Report · Decision Letter 1]

18 May 2023

COVID-19 vaccine uptake, confidence and hesitancy in rural KwaZulu-Natal, South Africa between April 2021 and April 2022: a continuous cross-sectional surveillance study

PGPH-D-23-00155R1

Dear Dr Guy Harling,

We are pleased to inform you that your manuscript 'COVID-19 vaccine uptake, confidence and hesitancy in rural KwaZulu-Natal, South Africa between April 2021 and April 2022: a continuous cross-sectional surveillance study' has been provisionally accepted for publication in PLOS Global Public Health.

Best regards,

Thu-Anh Nguyen

Academic Editor